# Can expected error costs justify testing a hypothesis at multiple alpha levels rather than searching for an elusive optimal alpha?

**Janet Aisbett** [ID]*

Complexity Science, Meraglim Holdings Corporation, Palm Beach Gardens, FL, United States of America

* janet.aisbett@meraglim.com

## Abstract

Simultaneous testing of one hypothesis at multiple alpha levels can be performed within a conventional Neyman-Pearson framework. This is achieved by treating the hypothesis as a family of hypotheses, each member of which explicitly concerns test level as well as effect size. Such testing encourages researchers to think about error rates and strength of evidence in both the statistical design and reporting stages of a study. Here, we show that these multi-alpha level tests can deliver acceptable expected total error costs. We first present formulas for expected error costs from single alpha and multiple alpha level tests, given prior probabilities of effect sizes that have either dichotomous or continuous distributions. Error costs are tied to decisions, with different decisions assumed for each of the potential outcomes in the multi-alpha level case. Expected total costs for tests at single and multiple alpha levels are then compared with optimal costs. This comparison highlights how sensitive optimization is to estimated error costs and to assumptions about prevalence. Testing at multiple default thresholds removes the need to formally identify decisions, or to model costs and prevalence as required in optimization approaches. Although total expected error costs with this approach will not be optimal, our results suggest they may be lower, on average, than when "optimal" test levels are based on mis-specified models.

## 1. Introduction

A long-standing debate concerns which, if any, alpha levels are appropriate to use as thresholds in statistical hypothesis testing, e.g., [1]. Among the issues raised is how the costs of errors should affect thresholds [2]. One approach is to determine optimal alpha levels based on these costs, or on broader decision costs, in a given research context.

When Type II error rates are presented as functions of the alpha level, a level can be selected that minimizes the sum of Type I and Type II error costs, contingent upon parameter settings such as effect and sample sizes. This is a local view of optimization. Optimization can also be interpreted in a global sense, as minimizing the expected total error cost given the prior probability of the hypothesis being true [3,4]. In either case, the total cost may also incorporate pay-offs from correct decisions [5,6].

**Competing interests:** The authors have declared that no competing interests exist.

In the global approach, estimates of the prior probability—also called the base rate or prevalence of true hypotheses—may be based on knowledge about the relationships under consideration, on replication rates in similar studies, or on other domain knowledge [7–9]. The prevalence estimate is then assumed to be the probability mass at the hypothesized effect size, with the remaining mass assigned according to the test hypothesis that the researcher hopes to reject—typically the null hypothesis of no effect. This dichotomous model has been extended to continuous distributions of true effect sizes in a research scenario [10,11].

Despite their mathematical and philosophical appeal, optimization strategies have not been widely adopted by researchers when selecting statistical test levels. This is arguably due to the difficulty of scoping, estimating, and justifying the various costs/payoffs [3], and of even defining a research scenario in which to estimate priors [12]. Greenland says that costs cannot be estimated without understanding the goals behind the analysis [2]. In view of the domain knowledge required, he suggests stakeholders other than researchers may be better placed to determine costs.

When studies involve well understood processes, domain knowledge will also guide estimation of prior probabilities of true hypotheses. Mayo & Morey argue, however, that such estimates require identifying a unique reference class, which is impossible, and, if it were possible, the prevalence proportion or probability would be unknowable [12].

And yet researchers are told they cannot hope to identify optimal alpha levels without good information about the base rate of true hypotheses and the cost of errors ([6]: S1 File). Relying on a standard threshold such as 0.05 is also not recommended, given the broad spread in optimal alpha values seen in plausible scenarios. For example, Miller & Ulrich find very small values are optimal when the prevalence of true hypotheses is low and the relative cost of Type I to Type II errors is high, and larger alphas are optimal when the converse holds [5,6].

## An alternative approach

We propose that, rather than try to find an optimum level in the face of such difficulties, researchers should simultaneously test at multiple test levels and report results in terms of these. This approach may deliver acceptable error costs without having to grapple with ill-defined notions of costs and research scenarios.

Simultaneous tests at multiple alpha levels lead to logical inconsistencies—when a hypothesis about a parameter value is rejected at one level but not rejected at another level, what can we say about the parameter value? This logical inconsistency can be overcome by extending the parameter space with an independent parameter that acts as a proxy for the test level [13]. The test hypothesis is extended with a conjecture about the value of the new parameter, so findings must also say something about it. This construction is simply to generate copies of the original hypothesis that can be identified by their proposed test level. The data are therefore extended so that all the added conjectures are rejected at their designated alpha level.

This construction puts testing at multiple alpha levels within the established field of multiple hypothesis testing. It is formalized mathematically in the S1 File, which also gives an example and shows how results from tests at multiple levels should be reported.

The advantage of such testing over simply reporting a raw P-value is that *a priori* attention must be paid to expected losses and the relative importance of Type I and Type II error rates and hence sample sizes. The advantage over a conventional Neyman-Pearson approach is that, at the study design stage, researchers have to consider more than one test level, and, at the reporting stage, they must tie findings to test levels.

We will refer to tests using this construction as *multi-alpha tests*. The remainder of the paper does not refer to the extended hypotheses and will speak of the original hypothesis being

rejected at one alpha level but not at another. This is shorthand for saying that the extended hypothesis linked to the first alpha level is rejected but the extended hypothesis linked to the second alpha level is not.

Error considerations in multi-alpha testing concern a family of hypotheses, even if the original research design only concerned one hypothesis. The family-wise error rate is the probability of one or more incorrect decisions. Obviously, family-wise error rates in multi-alpha testing are driven by the worst performers, namely Type I errors due to tests at the least stringent levels and Type II errors at the most stringent levels.

Looking at rates is misleading when alpha levels are explicitly reported alongside findings, because the costs of errors may vary with level. Many journals require that P-values be reported, since different values have different evidential meaning [14] and may lead to different practical outcomes. For example, when two portfolios are found to outperform the S&P 500 with respective P-values of 0.10 and 0.01, an investment web site [15] advises that "the investor can be much more confident" of the second portfolio and so might invest more in it.

Thus, rather than a statistical test leading to dichotomous decisions, there may be a range of decisions, each associated with different error costs. A small P-value may be interpreted as strong evidence that triggers a decision bringing a large positive payoff when the finding is correct and a large loss when the finding is incorrect as compared with the payoff when no evidence level is reported. Likewise, finding an effect only at a relatively weak level may lower the cost/payoff. As a result, total costs could on average be lower when hypotheses are tested at multiple alpha levels than when a single default such as 0.05 or 0.005 is applied over many studies.

Below, we generalize error cost calculations to scenarios with continuous effect size distributions and to multi-alpha testing. Examples based on the resulting formulas then compare expected total error costs from multi-alpha tests with costs from tests at single alpha levels, including optimal levels. Our findings support previous conclusions about the difficulty of choosing optimal alpha levels and they again illustrate how an alpha level that under some conditions is close to optimal can, with different prevalence and cost assumptions, bring comparatively high costs.

Testing the same hypothesis at two or more alpha levels smooths expected total error costs. The costs will not be optimal but may be lower on average than costs from tests using "optimal" alpha levels determined using inappropriate models.

## 2. Expected total cost of errors in a research scenario

We review and generalize conventional cost calculations to cover continuous effect size distributions. Then we adapt cost considerations to deal with multi-alpha tests and show that, in important cases, the expected error costs are a weighted sum of the expected costs of the individual tests.

To illustrate the mathematical formulas, and to highlight issues with their application, we draw on studies in which the subjects are overweight pet dogs and the study purpose is to investigate diets designed to promote weight loss. The primary outcome measure is percentage weight loss. Examples of such research are a large international study into high fiber/high protein diets [16], a comparison of high fiber/high protein with high fiber/low fat diets [17], and a study into the effects of enriching a diet with fish oil and soy germ meal [18]. Numerous published studies into nutritional effects on the weight of companion animals provide the basis for a relevant research scenario. Studies are frequently funded by pet food manufacturers that advertise their products as being based on expert scientific knowledge, as is the case with the studies cited above.

## Expected total error costs when testing at a single alpha level

**Dichotomous scenario.** Given a research scenario, suppose that proportion P of research hypotheses are true (or, for local optimization, set P = 0.5). Suppose "true" and "false" effect sizes are dichotomously distributed, with d the difference between the two values in a particular study. In our example, P is the proportion of dietary interventions on overweight pets that lead to meaningful weight loss and d is the standardized mean difference in weight loss.

When testing is at alpha level $\alpha$, the expected Type I error rate over many studies is $(1-P)\alpha$. The Type II error rate for the study can be expressed as a function $\beta(d, \alpha)$ with $P\beta(d, \alpha)$ the expected Type II error rate. The function $\beta$ will depend on the study design and on additional parameters such as sample size. In the study reported by Fritsch et al [17], $\beta$ is the chance that the difference in weight loss between the groups is not statistically significant at the study's test level 0.05 when the difference $d$ is meaningful. The trial had 30 dogs in the high protein group and 32 in the low-fat group, so in this case, $\beta(d, \alpha) = 1 - T(d/\sqrt{(1/30 + 1/32)} - T^{-1}(0.95))$ where $T$ is the cumulative $t$-distribution function with 60 degrees of freedom.

Suppose the cost of a Type I error is $C_0$ and the cost (or loss of benefit) of a Type II error is $C_1$, where these costs may be functions of the difference $d$. Then the expected total error cost, call it $\omega(d, \alpha)$, is

$$\omega(d, \alpha) = C_0(1 - P)\alpha + C_1 P\beta(d, \alpha). \tag{1}$$

The alpha level that minimizes this cost can be estimated by searching over alpha levels and sample sizes, given prevalence $P$, effect difference $d$ and relative error cost $C_0/C_1$ [3]. If optimization is against error rates rather than costs, this relative cost is fixed at 1.

In our example, a Type I error might lead veterinarians and dog food manufacturers to promote high fiber/low fat diet for weight loss when a high fiber/high protein diet would work as well and possibly have fewer adverse effects. Conversely, a Type II error might mean that a cheap and easily implemented weight loss approach was overlooked, with possibly less effective or more expensive high protein diets being promoted. Understanding the relative costs of these errors obviously requires expert knowledge of canine diet, obesity issues, weight-loss alternatives and so on. The costs also depend on how large an effect the low-fat diet has on canine weight compared with the other diet.

**Continuous scenario.** Suppose a histogram could be formed from the true size of effects of interventions targeting weight loss in companion animals. This histogram would approximate the density function that describes the prevalence of effect sizes in the research scenario.

More generally, let $p(e)$ describe the prevalence of effect size $e$ in a domain $R$ (typically, an interval). Let $E$ be the subset of effect sizes which satisfy the test hypothesis that the researcher is seeking to reject. $R–E$ is therefore the subset of effects for which the research hypothesis is true. For example, if veterinarians consider a standardized weight loss of more than 0.1 to be practically meaningful, and the researcher wants to show a diet leads to meaningful weight loss, $R–E$ is anything larger than 0.1.

The Type I error rate is now also a function of effect size, since when true effects are further away from $R–E$—for example, if the dogs on the diet put on weight so the standardized weight loss is negative—trials are less likely to incorrectly reject the test hypothesis. Denote the Type I error rate as $\beta_0(e, \alpha)$. Then $\beta_0(e, \alpha) \leq \alpha$ and $\int_{e \in E} \beta_0(e, \alpha)p(e)\mathrm{de}$ is the average Type I error rate (assuming that the functions are integrable). Likewise, if the Type II error rate is $\beta_1(e, \alpha)$, the average Type II error rate is $\int_{e \in R-E} \beta_1(e, \alpha)p(e)\mathrm{de}$. These integrals respectively replace $(1-P)\alpha$ and $P\beta(d, \alpha)$ in (1) when modelling a continuous prevalence scenario. If costs and benefits are

also functions of effect size, the expected total error cost $\varpi(p, \alpha)$ in the continuous scenario is

$$\varpi(p, \alpha) = \int_{e \in E} C_0(e)\beta_0(e, \alpha)p(e)de + \int_{e \in R-E} C_1(e)\beta_1(e, \alpha)p(e)de. \qquad (2)$$

Although the assumption of a continuous prevalence distribution is appealing, estimating the distribution is potentially even more difficult than estimating a prevalence proportion $P$ in the dichotomous modelling. Any prevalence model compiled from relevant published literature will be distorted by publication bias and by the discrepancies between reported and true effect sizes highlighted by the "reproducibility crisis".

## Expected total error costs when testing one hypothesis at multiple alpha levels (multi-alpha testing)

**Decisions, costs, and test levels.**   Greenland argues that decisions cannot be justified without explicit costs [2]. We therefore relate decisions to costs and test levels before extending the cost expressions to multi-alpha testing.

Consider a hypothesis, such as that higher fiber diets increase weight loss in overweight dogs or that an intervention to control an invasive plant species is more efficacious than a standard treatment under some specified conditions. Suppose a set of decisions D(m), m = 0, 1, 2, . . ., k, concerns potential actions to be taken, with D (0) the decision to take no new action.

For instance, decisions might be directly research-related, such as terms used in reporting the study findings or choices of publication vehicle. Or they might concern practical actions, such as the study team encouraging local veterinarians to recommend low fat diets over high protein diets, or the research sponsor publicizing the benefits of switching to low fat diets, or the sponsor manufacturing low fat products and promoting them as better than high protein products.

The decisions are ordered according to their anticipated payoff $C_1(m)$ compared with the decision $D(0)$ to take no new action, given that the intervention has a meaningful effect. Thus, $D(k)$ will bring the greatest payoff $C_1(k)$. For example, publication in a more highly ranked journal may improve the researchers' resumés. In the canine example, the overall health benefit from weight loss in overweight dogs increases with the number of dogs that switch from high protein to low fat diets; thus the research sponsor offering a low fat product as well as advertising the benefits will have greater payoff than the other actions.

However, deciding $D(m)$ when the effect is not meaningful brings a cost, call it $C_0(m)$. These costs are assumed to increase with level $m$ so that decision $D(k)$ carries the greatest cost if the intervention has no meaningful effect. This cost might include more critical attention if published findings are not supported by later studies. In the canine example, costs include promotional costs, the continuing health costs for overweight dogs which might have otherwise had a more effective high protein diet, and potential impacts on the researchers' and sponsor's reputations.

Finally, assume that a decreasing set of alpha levels $\alpha_m$, m = 1, . . ., $k$ has been identified, such that decision $D(m)$ will be made if the test hypothesis is rejected at alpha level $\alpha_m$ but not at level $\alpha_{m+1}$. That is, only one decision is to be made, and it will be selected according to the most stringent alpha level at which the test hypothesis is rejected. If the test is not rejected at any of the alphas, then the decision is $D$ (0), do nothing.

Thus, the research team and sponsor might decide to encourage local veterinarians to promote low fat diets over high protein ones if the findings of the canine dietary study are statistically significant at alpha level 0.05; to recommend such diets on the sponsor's website and to

veterinarian societies if findings are significant at level 0.01; and to manufacture low fat products and promote switching to them if findings are significant at level 0.001.

One approach to making such choices is provided at the end of this section, and the Discussion section further looks at how such a set of alpha levels might be identified.

**Expected total costs of errors.** Suppose P is the prevalence of true hypotheses in a research scenario in which a dichotomous distribution of effects is assumed (or if error costs are considered locally, set P = 0.5).

Eq (1) is the expected error cost when a test will result either in a decision involving an action, or in doing nothing, with the action taken when the P-value calculated from the data is below $\alpha$. In the multiple alpha level testing scenario, however, decision $D(m)$ is only made when the P-value lies between $\alpha_{m+1}$ and $\alpha_m$. For example, local veterinarians would only be enlisted by the canine diet researchers if the findings are statistically significant at level 0.05 but are not at level 0.01, since at the higher level a wider campaign would be undertaken involving veterinarian organizations.

The expected total Type I error cost, call it $\omega_0$, over all the possible decisions is therefore

$$\omega_0 = (1 - P)\sum\nolimits_{m=1,\ldots,k}C_0(m)(\alpha_m - \alpha_{m+1}) = (1 - P)\sum\nolimits_{m=1,\ldots,k}(C_0(m) - C_0(m - 1))\alpha_m. \quad (3)$$

Here, we set $\alpha_{k+1} = 0 = C_0(0)$, so that $\sum\nolimits_{m=1,\ldots,k}C_0(m)\alpha_{m+1} = \sum\nolimits_{m=1,\ldots,k-1}C_0(m)\alpha_{m+1}$

$$\equiv \sum\nolimits_{m=2,\ldots,k}C_0(m - 1)\alpha_m = \sum\nolimits_{m=1,\ldots,k}C_0(m - 1)\alpha_m,$$

and then rearranged terms to get the second equality.

When the effect of an intervention is meaningful, but the test hypothesis is not rejected by a test at some alpha level, the error cost depends on the difference between the largest payoff $C_1(k)$ and the payoff brought by what *was* decided. That is, if the decision is $D(m)$, the loss is $C_1(k) - C_1(m)$. In our example, if low fat canine diets are more effective that high protein diets yet the decision was to only promote them through local veterinarians, the loss would be due to the benefit difference between that and the wider advertising campaign and offering of products that saw more overweight dogs put on such diets.

If the decision is $D(k)$, there is no loss, in the sense that the strongest of the decision choices has been made. If no test rejects the test hypothesis, the decision is $D(0)$ and the loss is $C_1(k)$. The total expected Type II error cost $\omega_1$ is therefore

$$\omega_1 = PC_1(k)\beta(\alpha_1) + P\sum\nolimits_{m=1,\ldots,k-1}(C_1(k) - C_1(m))(\beta(\alpha_{m+1}) - \beta(\alpha_m))$$

$$= PC_1(k)\beta(\alpha_k) - P\sum\nolimits_{m=1,\ldots,k-1}C_1(m)(\beta(\alpha_{m+1}) - \beta(\alpha_m)) \quad (4)$$

where, to simplify notation, the dependence of the Type II error rate $\beta$ on effect size is suppressed.

Rearranging terms and setting $C_1(0) = 0$ yields

$$\omega_1 = P\sum\nolimits_{m=1,\ldots,k}(C_1(m) - C_1(m - 1))\beta(\alpha_m). \quad (5)$$

The expected total error cost $\omega = \omega_0 + \omega_1$ from testing at multiple alpha levels is therefore

$$\omega = \sum\nolimits_{m=1,\ldots,k}((1 - P)\Delta C_0(m)\alpha_m + P\Delta C_1(m)\beta(\alpha_m)), \quad (6)$$

where $\Delta C_0(m) = C_0(m) - C_0(m - 1)$, $\Delta C_1(m) = C_1(m) - C_1(m - 1)$. This is the sum in (6)

is the sum of expected total error cost over all the decisions when costs are replaced by the differences between adjacent costs.

When the distribution of effects in the research scenario is modelled by a density function $p(e)$ and costs are dependent on effect size, expression (6) for total expected cost becomes

$$\sum_{m=1,\dots,k} \left( \int_{e\in E} \Delta C_0(m;e)\beta_0(e,\alpha_m)p(e)de + \int_{e\in R-E} \Delta C_1(m;e)\beta_1(e,\alpha_m)p(e)de \right). \qquad (7)$$

Here, $\Delta C_0(m;e)$ is the difference in costs for decisions $D(m)$ and $D(m-1)$ when the effect size is $e \in E$, and $\Delta C_1(m;e)$ is the difference between payoffs when $e \in R-E$.

## Expected error cost of a multi-alpha test as the weighted sum of the costs of the single level tests

In standard single-level test situations within a conventional Neyman-Pearson framework, the potential research outcomes are independent of test level, according to the "all or nothing" nature of findings when a test threshold is applied. These dichotomous outcomes can plausibly be set to $D(0)$ (do nothing if the test hypothesis is not rejected) and $D(k)$. Thus, if the canine diet study gets a significant result, the sponsor will go ahead with manufacturing and marketing a product; otherwise, say, the researchers and the sponsor's product innovation team may be asked to identify factors affecting the result. The Type I and Type II costs for these decisions are $C_0(k)$ and $C_1(k)$ respectively.

Now suppose that the relative cost $r$ of Type II to Type I errors is the same at each test level, so that $C_1(m) = rC_0(m)$ for $m = 1, \dots, k$. For example, if both cost types are proportional to the number of dogs switching to a low fat diet on the basis of a study, and decisions made at the different test levels primarily determine this number, then the ratio would be approximately constant.

In such a case, it is straightforward to show (see S2 File) that the total cost of the multi-alpha test is a weighted average of the costs of the individual tests given in (1). It follows that the multi-alpha test will always be less costly than the worst, and more costly than the best, of the single level tests. It is thus more costly than the optimal single level test. The case with a continuous distribution of true effects is analogous.

## The value of the information obtained from a test

What might be said about the relationship between costs at different test levels? Rafi & Greenland [19] suggest that the information conveyed by a test with P-value $p$ is represented by the surprisal value $-\log_2(p)$. The smaller the $p$, the more informative the test. In a multi-alpha test, the information conveyed on rejecting a test hypothesis at level $\alpha_m$ might therefore be proportional to $-\log_2(\alpha_m)$.

If each decision $D(m)$ is logically linked to the (correct or incorrect) information provided by a test at level $\alpha_m$, its associated costs would then also be proportional to $-\log_2(\alpha_m)$. That is, for some constants $C, C'$,

$$C_0(m) = -C\log_2(\alpha_m), C_1(m) = -C'\log_2(\alpha_m) \equiv \left(\frac{C'}{C}\right)C_0(m), m = 1, \dots, k. \qquad (8)$$

Therefore, when error costs at each test level are proportional to the surprisal value of that level, the total cost of the multi-alpha test is a weighted average of the costs of the component tests. This analysis carries over to the case of a continuous distribution of true effect sizes.

The S2 File has mathematical details. It also shows how the first equality in (8) can be used to set alpha levels if costs are known, or conversely can be used to guide appropriate decisions given pre-set alpha levels.

The surprisal values of test levels 0.05, 0.01 and 0.001 are respectively 4.3, 6.6 and 10.0. The information brought by the test at level 0.01 is therefore 50% more than that of the test at 0.05, and that brought by the stringent test at level 0.001 is 50% larger again. If experts agree that the error costs associated with the hypothetical decisions in our low fat versus high protein canine diet study increase by much more than 50% between levels, the range of test levels would need to be increased.

Note that transforming P-values to information values assumes no prior knowledge. A small P-value for a non-inferiority test contrasting the effectiveness of drug A with drug B would not be "surprising" if drug A had been shown to be superior in many previous trials. Indeed, over a very large sample, a large P-value would be surprising in this case.

## 3. Applying the expected cost formulations

We compare expected error costs of multi-alpha tests with those from conventional testing in studies investigating whether an effect size is practically meaningful. This offers insights into the cost behavior of multi-alpha tests, as well as illustrating the sensitivity of optimization and the potential pitfalls of using a single default test level. Research scenarios are modelled using both dichotomous and normally distributed true effect sizes, and costs from tests at multiple alpha levels are compared with those from single level tests and tests at optimal levels.

The first subsection uses cost estimates drawn from a simplified example in which Type II error costs vary with effect size and can be much higher than Type I costs. The second subsection investigates a research scenario in which different research teams anticipate different effect sizes, and so conduct trials with different sample sizes. Both true effect sizes and anticipated effect sizes are assumed to be normally distributed, and costs are reported as the total expected costs averaged over all the research teams.

We apply test alpha levels ranging from extremely weak through to strong to highlight how each of the levels can give a lower expected total error cost than the others under some parameter settings. Throughout, a two-group design with equal groups is assumed, where $n$ is the total sample size, $M$ is the boundary between meaningful and non-meaningful effect sizes, and all alpha levels are with respect to one-sided tests. For the multi-alpha tests, error costs are assumed to be proportional to the surprisal value of the component test level. This simplifying assumption implies no prior knowledge about test outcomes.

S3 File further illustrates the sensitivity of cost computations and the smoothing effect that multi-alpha testing has on error rates. It summarizes results from simulations in which cost differences between test levels are randomly assigned but Type I errors are on average more costly. The simulations apply one-sided test levels commonly found in the literature.

### Cost comparisons in an example research scenario

Consider a research scenario of clinical trials investigating Molnupiravir as therapy in non-hospitalized patients with mild to moderate symptoms of Covid 19. Suppose the primary outcome is all-cause hospitalization or death by day 29.

Following Ruggeri et al [20] and Goswami et al [21], only consider the economic burden when estimating costs. If an ineffective drug is administered, economic costs stem from the direct price of the drug and from its distribution. If a drug that would reduce hospitalizations is not used, the economic costs stem from hospitalizations and deaths that were not avoided;

the lower the risk ratio, the more hospitalizations could have been prevented using the drug, and hence the greater the costs of Type II errors.

Given a study of Molnupiravir in a particular population, suppose the true risk of hospitalization or death is $r_1$ in the untreated group and $r_2$ in the treatment group. Let $I$ be the risk of getting mild to moderate Covid in the population of interest. If the average cost of administering Molnupiravir to an individual is $c_T$ then, over the population, the per capita cost of administering it to those diagnosed with mild to moderate Covid is $C_0 = c_T I$.

Without treatment, the chance of getting Covid and then being hospitalized or dying within 29 days is $I r_1$ while this chance when Molnupiravir is administered is $I r_2$. If $c_H$ is the average cost of a hospitalization episode or death due to Covid then the per capita saving/loss from using the drug is $c_H I (r_1 - r_2)$. Therefore, the cost of not using the drug if it is effective ($r_1 > r_2$) is the difference between this cost and the cost of administering it:

$$C_1 = c_H I (r_1 - r_2) - c_T I. \tag{9}$$

The difference can be negative. The drug would therefore only be economical to administer if the absolute risk reduction exceeds $c_T/c_H$, which in this simplified example we take to be the definition of practically meaningful.

A Molnupiravir cost-benefit analysis using US data [21] set the cost of a treatment course with the drug at \$707 and the cost of a Covid hospitalization (without ICU) at \$40,000, with the risk of hospitalization for the untreated population about 0.092. On these conservative notions of cost, the risk must be reduced by at least $707/40{,}000 \approx 0.018$ for the drug rollout to be cost-effective. These values are used in the calculations reported below.

### Dichotomous distribution of effects in the research scenario

Researchers are testing the null hypothesis that the risk difference $r_2 - r_1$ exceeds the break-even point $M = -c_T/c_H$. Suppose that the proportion of Molnupiravir trials involving meaningful effects is $P$ and that, in the current study, the true risk difference $r_2 - r_1 < M$. The expected critical value when testing against a risk difference $M$ at level $\alpha$ is approximately $M + s\, z_\alpha$, where $s = \sqrt{2(r_1(1 - r_1) + (r_1 + M)(1 - r_1 - M))/n}$. Set $d = r_2 - r_1 - M$. The expected Type II error rate $\beta(d, \alpha)$ then satisfies

$$z_{1-\beta} = (-d + s\, z_\alpha)/ss \tag{10}$$

where $ss = \sqrt{(2(r_1(1 - r_1) + r_2(1 - r_2))/n)}$.

Table 1 reports expected total error costs for various parameter settings calculated from (1) with $\beta(d, \alpha)$ and costs defined as above (dropping the scale factor $I$ which appears in all terms).

**Table 1. Expected total error costs per Covid patient on testing risk difference (RD) against break-even difference M in dichotomous research scenarios.**

|  | $\alpha_1 = 0.25$ | $\alpha_2 = 0.05$ | $\alpha_3 = 0.001$ | *multi-level* | *optimal* |
|---|---|---|---|---|---|
| *RD = -0.025* |  |  |  |  |  |
| *P = 0.5* | 166.5 | **143.5** | 146.1 | 149.5 | 143.2 (0.04) |
| *P = 0.1* | 174.7 | 57.0 | **29.8** | 65.2 | 29.3 (0.00) |
| *RD = -0.05* |  |  |  |  |  |
| *P = 0.5* | **98.4** | 108.1 | 452.5 | 301.2 | 77.5 (0.13) |
| *P = 0.1* | 161.1 | **49.9** | 91.1 | 95.5 | 45.8 (0.03) |

P is the proportion of true effects in the research scenario. The lowest of the costs of the single level tests is shown in bold. The one-sided alpha level at which expected total costs are minimized is in brackets in the final column. Calculations assume two groups of 1000.

Single level tests are at the one-sided alphas shown in columns 2–4, and multi-alpha tests are formed from these three levels. Note that group size 1000 gives *a prior* calculated power of 80% to detect effect size –0.046 if testing against the break-even point $M$ at alpha level 0.05, assuming risk in the untreated group is 0.092.

The table reveals a range of optimal alpha levels, allowing each the selected single test levels to out-perform the others in some setting. The values $P = 0.5$ and $P = 0.1$ respectively model the local or "no information on prevalence" case and the pessimistic research scenario [8]. If the true risk difference is near the break-even point and prevalence is low, Type I errors dominate and small alphas are optimal. For larger risk differences or higher prevalence, larger alphas help limit the more costly Type II errors.

For the multi-alpha tests, costs are assumed to be proportional to the surprisal value of the component test level, as described earlier. Proportionality could result from deciding only to administer Molnupiravir to a proportion of people with Covid symptoms, for example, limiting the roll-out to some locations, where the proportion depends on the test level at which the hypothesis of no effect is to be rejected. Then, as discussed earlier, the total expected cost of the multi-alpha tests is a weighted sum of the total expected costs of the one-sided tests at each level. Because of the wide range of alphas, these weighted averages can be substantially higher than the optimal costs. We will return to this point in the Discussion.

**Continuous distribution of effects in the research scenario.**  Suppose the risk difference between treatment and control groups in the Molnupiravir studies approximately follows a normal distribution, and that in a study the true risk in the untreated group is $r_1$.

Given critical value $M + s\, z_\alpha$ and standard deviations $s$ and $ss$ defined as above, the standard normal cumulative probability at $(r_1 - r_2 + M + s\, z_\alpha)/ss$ is the expected Type I error rate for effect sizes $r_2 - r_1$ larger than $M$ and is the expected power for effect sizes smaller than $M$. This formulation equates to that given by (10) for calculating expected Type II error rates.

Based on these calculations, Table 2 reports expected total error costs for different research scenario distributions. For the multi-alpha tests, costs are again assumed to be proportional to the surprisal value of the test level, so that the total expected cost is a weighted sum of the single level test costs. This table is striking for the very large alphas it shows optimizing expected total error costs. The wide range of alpha levels involved in the multi-alpha tests again lead to costs that can be far from optimal.

A normal distribution of true effects about –0.02 is an optimistic research scenario, in that interventions have more than 50% chance of being meaningful (i.e., cost effective). A normal distribution about zero with a tight standard deviation of 0.015 represents a pessimistic scenario, with only 12% chance of meaningful intervention. The test level 0.05 is close to optimal in the pessimistic scenario because Type II errors are rare. Costs are also lower for each test level in this scenario compared with the optimistic scenario because of the low probability of costly Type II errors. This is illustrated graphically in the S4 File.

**Table 2.  Expected total error costs in research scenarios where true risk differences follow a normal distribution with mean and standard deviation shown in the first column.**

|  | $\alpha_1 = 0.25$ | $\alpha_2 = 0.05$ | $\alpha_3 = 0.001$ | *multi-level* | *optimal* |
|---|---|---|---|---|---|
| $\mu = 0$ |  |  |  |  |  |
| $\sigma = 0.015$ | 39.9 | **28.4** | 33.9 | 33.8 | *28.3 (0.06)* |
| $\sigma = 0.025$ | **45.3** | 62.1 | 109.6 | 85.6 | *45.1 (0.23)* |
| $\mu = -0.02$ |  |  |  |  |  |
| $\sigma = 0.015$ | **96.3** | 154.4 | 254.3 | 199.3 | *93.0 (0.34)* |
| $\sigma = 0.025$ | **69.9** | 130.8 | 281.0 | 203.7 | *65.7 (0.36)* |

Note that when Type II errors have higher cost, Neymann advises the test direction should be reversed because Type I errors rates are better controlled [22]. In this example, decisions would then be made according to the rejection level, if any, of the tests of the hypothesis that Molnupiravir treatment *was* cost-effective.

### A research scenario in which different research teams anticipate different standardized effect sizes

Consider a research scenario in which studies collect evidence about whether an intervention is practically meaningful, in the sense of a standardized effect difference exceeding some boundary value *M*.

In the study design stages, different research teams make different predictions about the effect size. For example, the core literature might support a value such as $M + 0.4$, say, but each research team may apply other evidence to adjust its prediction. Suppose these predictions approximately follow a distribution with density function $p'(x)$.

Given an anticipated effect size, each research team calculates sample sizes to achieve 80% power at a one-sided test level of 0.025, using the standard formula. For simplicity, suppose each team selects equal sized groups and estimates the same error costs. Finally, assume the true standardized effects in the research scenario follow a continuous distribution with density function $p(x)$.

Fig 1A illustrates distributions of true and anticipated effect sizes. Fig 1B shows how the research teams' sample sizes vary as a function of the effect sizes they anticipate. Fig 1C converts Fig 1B into a density function describing the probability that a sample size is selected in the research scenario. The requirement to achieve 80% power means that teams who are pessimistic about the anticipated effect size may need very large samples.

The differing sample sizes in the studies affect both Type I and Type II error rates, denoted $\beta_0$ and $\beta_1$ in Eq (2). These error rates, and hence the expected total error cost $\varpi(p, \alpha)$, are functions of the anticipated effect size *x*. The expected total cost over all studies at a single test level $\alpha$ is then $\int \varpi(p, \alpha) p'(x) dx$, for $\varpi(p, \alpha)$ as defined in Eq (2). The multi-alpha expected total cost over all studies is similarly obtained by integrating the expression in (7).

Table 3 reports expected total costs averaged over all studies, for different means of the true effect distribution and different ratios of Type I to Type II error costs. The costs for the tests at the more stringent level hardly vary with cost ratio because the Type I error rate is negligible. The higher Type I error rates for the weak level tests cause costs to decrease with smaller cost ratios.

Averaged expected costs for the multi-alpha test are intermediate between the costs of the single level tests and can thus be seen as smoothing costs compared with testing at either level as a default. However, the optimal test levels can be even more lenient than we saw in Table 2, making the optimal average expected costs substantially lower than for the other tests reported in Table 3.

## 4. Discussion

A single alpha level cannot be appropriate for all test situations. Yet it is difficult to establish the level at which costs will be approximately minimized in any given research context. As Miller & Ulrich noted [6], and as our examples show, optimization is very sensitive to the proportion of true test hypotheses in the research scenario. Even allowing for perfect knowledge of the various costs and of the nature of the distribution of true effect, very different alpha levels can be close to optimal with different sample sizes and different parameters of the effect size distribution.

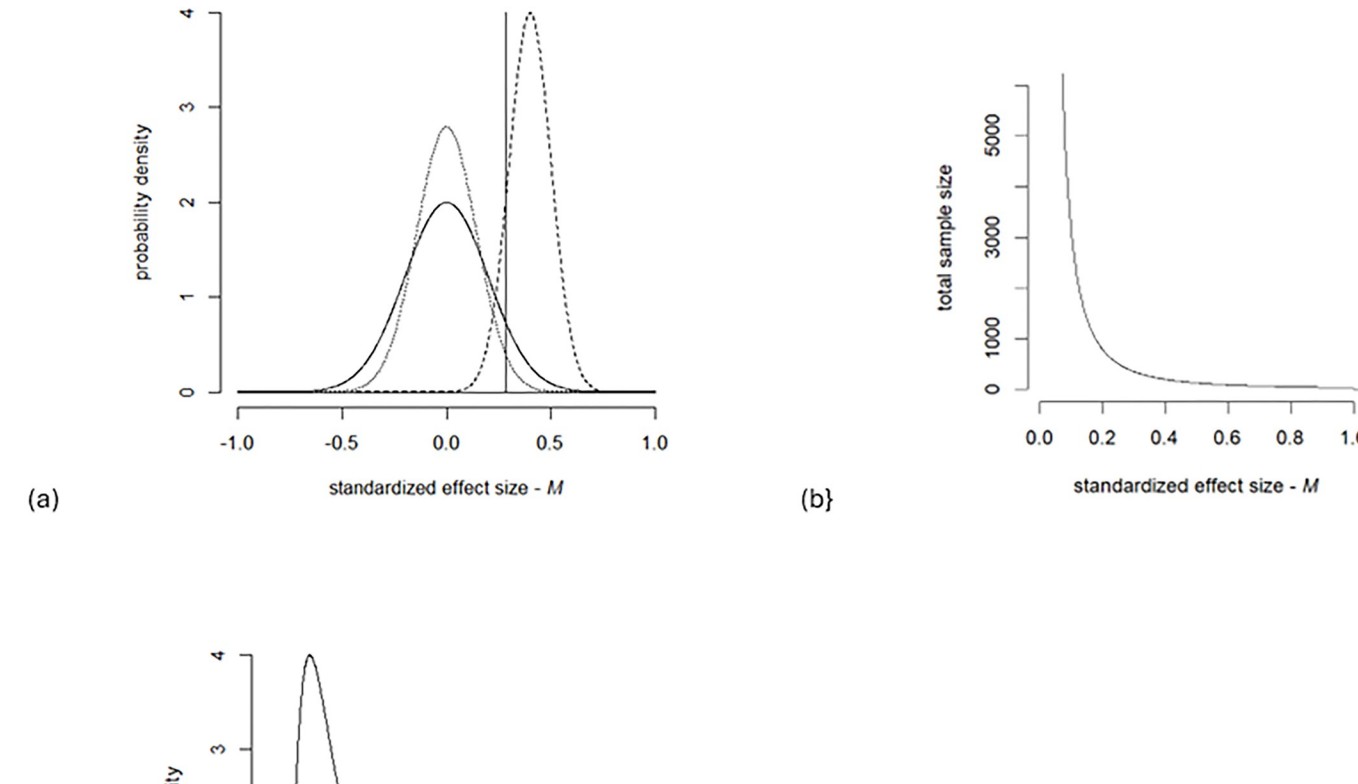

**Fig 1. Research scenario in which both true and anticipated standardized effect sizes are normally distributed.** (a) Distributions of true (solid curve) and anticipated (dashed curve) standardized effect sizes, assuming true effects are centered on M. The dotted curve is the sampling distribution of an effect on the boundary of meaningful effects when total sample size is 98. The heavy vertical line is at the critical value for level 0.025 tests on this sample, indicating a high Type II error rate. (b) Sample sizes as function of anticipated effect size, calculated using normal distribution approximations with α = 0.025 and 80% power. (c) Probability distribution of sample sizes in the research scenario given the distribution of anticipated effect sizes shown in (a).

The impact of the research scenario is not surprising, given how true effect size distributions weight expected Type I versus Type II error rates and hence weight relative costs. Previous investigations into optimal test levels modelled effect sizes as taking one of two possible values. We followed [11] in also investigating research scenarios in which effect sizes are continuously distributed.

We accepted the convention that rejection of the test hypothesis leads to stakeholders "acting as if" the alternate hypothesis is true, and hence assumed that costs in single level tests are independent of the test level. In the multi-alpha tests, findings are tied to their test level. The costs of a false rejection were therefore assumed to be lower if tests at more stringent levels were not rejected, since it is not plausible to "act as if" in the same way. We suggested that the

**Table 3. Expected total error costs averaged over a research scenario in which anticipated standardized effect sizes are normally distributed as N(M+0.4, .01).**

|  | *α = 0.25* | *α = 0.025* | *multi-level* | *optimal* |
|---|---|---|---|---|
| *cost ratio = 10* |  |  |  |  |
| *true mean = M—0.1* | 0.44 | **0.24** | 0.32 | 0.16 (0.02) |
| *true mean = M* | 0.47 | **0.34** | 0.39 | 0.32 (0.06) |
| *true mean = M + 0.4* | **0.11** | 0.24 | 0.19 | 0.06 (0.36) |
| *cost ratio = 4* |  |  |  |  |
| *true mean = M—0.1* | 0.24 | **0.22** | 0.23 | 0.14 (0.02) |
| *true mean = M* | **0.28** | 0.32 | 0.31 | 0.27 (0.16) |
| *true mean = M + 0.4* | **0.09** | 0.24 | 0.18 | 0.03 (0.36) |
| *cost ratio = 1* |  |  |  |  |
| *true mean = M—0.1* | **0.15** | 0.21 | 0.19 | 0.09 (0.44) |
| *true mean = M* | **0.19** | 0.31 | 0.26 | 0.08 (0.43) |
| *true mean = M + 0.4* | **0.08** | 0.24 | 0.18 | 0.01 (0.36) |

*Here* true mean *is the mean of the normal distribution of true effect sizes, assumed to have standard deviation* 0.2. *M is the smallest meaningful effect size.*

costs at each alpha level might be proportional to the surprisal value (also called information content) of a finding with a P-value at that level. Our reasoning was that, with less information, any response would be more muted and therefore Type I costs would be lower. On the other hand, Type II costs would be lowered when the test hypothesis was rejected at some, but not all, the test levels, because a little information is better than none.

Research is needed into how reporting findings against test levels affects their interpretation, and hence affects costs. This is related to how reporting P-values affects decisions, although multi-alpha test reporting goes further, in explicitly stating, for example, that a study did not provide any information about the efficacy of an intervention at level 0.005 but did at level 0.05. The results presented in section 3 and in the S3 File obviously give just a glimpse into how expected total error costs vary with modelling assumptions. Nevertheless, the results suggest costs from testing at multiple alpha levels are smoothed from the extremes of the costs when tests in different research scenarios are made at one fixed alpha level. Testing at an "optimal" level calculated using models based on incorrect assumptions may also lead to larger costs than testing at multiple levels.

Theoretically, multiple test levels could be chosen to optimize expected total costs, using Eq (6), say, and searching over the multidimensional space formed by sample size and the vector of alpha levels. The problems of estimating unknowables would obviously be worse, given our cost formulations assign different error costs at each of the levels. Aisbett [13]: Appendix 2 adapts an optimization strategy to the multi-alpha case in part by making simple assumptions about cost behavior. However, we do not recommend trying to optimize error costs across tests at multiple alphas.

How then should the alpha levels be selected? When an alpha that incurs high costs is included in the set of test levels, the multi-alpha test costs are increased. For example, in the low prevalence conditions in Table 1, the high Type I error rates from testing at 0.25 blows out the difference between the optimal or best performer costs and those of the multi-alpha test.

With this in mind, we suggest three strategies for setting the alpha levels in a multi-alpha test.

The first approach is to calculate optimal alphas for single level tests against a range of cost and prevalence models that are reasonable for the research scenario, and then use the smallest

and largest of these alphas. The expected error costs from the multi-alpha tests will smooth out high costs if incorrect models are chosen.

A second strategy appropriate for applied research is to follow the procedure presented, without justification, in section 2. In this, potential decisions/courses of actions are identified, associated costs are estimated in terms of the consequences of incorrect decisions, and then test alpha levels are assigned. We envisage this to be an informal process rather than a search for a mathematically optimal solution. Greenland [2] recommends that researchers identify potential stakeholders who may have competing interests and hence differing views on costs, as Neyman illustrated with manufacturers and users of a possibly carcinogenic chemical [22]. Testing at multiple alpha levels selected *a priori* may allow these differing views to be incorporated at the study design stage. In the reporting stage too, different stakeholders can act differently on the findings according to their perceptions of costs, since findings are tied to test levels and weaker levels are associated with lower costs.

A third approach that avoids estimating error costs is the pragmatic strategy of assigning default values, guided by common practice in the researcher's disciplinary area. Common practice may also be incorporated in standards such as those of the U.S. Census Bureau which define "weak", "moderate", and "strong" evidence categories of findings through the test levels 0.10, 0.05 and 0.01 [23]. Default alpha levels might also be assigned indirectly using default effect sizes, such as the Cohen's d values for "small", "medium" and "large" effects, given the functional relationships between P-values and effect sizes that depend, inter alia, on sample size and estimated variance. This pragmatic approach is open to the criticisms made about default alphas in the single level case, but, again, it smooths out high costs that arise when a default alpha level is far from optimal.

An appealing variation on this approach builds on links between Bayes factors (BFs) and P-values [4]. BFs measure the relative evidence provided by the data for the test and alternate hypotheses. Formulas relating P-values to BFs have been provided for various tests and assumed priors [24]. Supporting software computes alpha levels that deliver a nominated BF for a given sample size, subject to the assumptions. Alpha levels in some multi-alpha tests could therefore be set from multiple BF nominations. Intervals on the BF scale are labelled "weak", "moderate" and "strong" categories of evidence favoring one hypothesis over the other. The boundaries of these intervals are potential default BF nominations. The corresponding alphas are functions of sample size, avoiding Lindley's Paradox when sample sizes are large.

Whatever levels are chosen, testing at multiple alphas is subject to the qualifications needed for any application of P-values, which only indicate the degree to which data are unusual *if* a test hypothesis together with all other modelling assumptions are correct. Data from a repeat of a trial may still, with the same assumptions and tests, yield a different conclusion.

Nevertheless, testing at more than one alpha level discourages dichotomous interpretations of findings and encourages researchers to move beyond p < 0.05. We have shown that, rather than raising total error costs, multi-alpha tests can be seen as a compromise, offering adequate rather than optimal performance. Such costs, however, may be lower than those from optimization approaches based on mis-specified models. Empirical studies involving committed practitioners are needed in diverse fields, such as health, ecology and management science, to better understand the relative cost performance and to evaluate practical strategies for setting test levels.

## Supporting information

**S1 File. Testing one hypothesis at multiple alpha levels: theoretical foundation and indicative reporting.**
(PDF)

**S2 File. Total cost of the multi-alpha test as a weighted average of the costs of the individual tests.**
(PDF)

**S3 File. Averaged results from simulations with random costs.**
(PDF)

**S4 File. Illustration of impact of the distribution of effect sizes in the research scenario on Type I and Type II error rates.**
(PDF)

**S5 File. R code to produce all Tables and Figures: See *github.com/JA090/ErrorCosts*.**
(DOCX)

## Author Contributions

**Conceptualization:** Janet Aisbett.

**Formal analysis:** Janet Aisbett.

**Methodology:** Janet Aisbett.

**Software:** Janet Aisbett.

**Writing – original draft:** Janet Aisbett.

**Writing – review & editing:** Janet Aisbett.

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
