## [Decision Letter · Decision Letter 0]

22 Jan 2024

PONE-D-23-36250An alternative to chasing an elusive alpha that minimizes ill-defined error costs: simultaneously test a hypothesis at multiple alpha levelsPLOS ONE

Dear Dr. Aisbett,

Thank you for submitting your manuscript to PLOS ONE. After careful consideration, we feel that it has merit but does not fully meet PLOS ONE’s publication criteria as it currently stands. Therefore, we invite you to submit a revised version of the manuscript that addresses the points raised during the review process.

**We have now received feedback from two expert reviewers regarding your manuscript. Both reviewers, along with my personal evaluation, find your paper to be promising. I request you to prepare a revised version of your paper, taking into consideration the comments provided by the reviewers. Their suggestions have the potential to significantly enhance the quality of your manuscript. In your revision, please pay special attention to the points raised about the complexity, practicality, and practical relevance of your results. Furthermore, the reviewers have emphasized the need for a comparative discussion of your approach against other existing methods. Elaborating on how your approach fits with the principles of significance testing as defined by Neyman and Pearson could greatly enrich your paper's content. Reviewer 1 has also mentioned the need for further empirical validation of your method in varied research settings. While this might be beyond the scope of the current paper, I suggest you include this comment in the section that critically discusses your approach and outlines potential directions for future research.** Please submit your revised manuscript by Mar 07 2024 11:59PM. If you will need more time than this to complete your revisions, please reply to this message or contact the journal office at plosone@plos.org. Please include the following items when submitting your revised manuscript:A rebuttal letter that responds to each point raised by the academic editor and reviewer(s). You should upload this letter as a separate file labeled 'Response to Reviewers'.A marked-up copy of your manuscript that highlights changes made to the original version. You should upload this as a separate file labeled 'Revised Manuscript with Track Changes'.An unmarked version of your revised paper without tracked changes. You should upload this as a separate file labeled 'Manuscript'.

We look forward to receiving your revised manuscript.

Kind regards,

Stephan Leitner

Academic Editor

PLOS ONE

Journal Requirements:

Additional Editor Comments:

We have now received feedback from two expert reviewers regarding your manuscript. Both reviewers, along with my personal evaluation, find your paper to be promising. I request you to prepare a revised version of your paper, taking into consideration the comments provided by the reviewers. Their suggestions have the potential to significantly enhance the quality of your manuscript. In your revision, please pay special attention to the points raised about the complexity, practicality, and practical relevance of your results. Furthermore, the reviewers have emphasized the need for a comparative discussion of your approach against other existing methods. Elaborating on how your approach fits with the principles of significance testing as defined by Neyman and Pearson could greatly enrich your paper's content. Reviewer 1 has also mentioned the need for further empirical validation of your method in varied research settings. While this might be beyond the scope of the current paper, I suggest you include this comment in the section that critically discusses your approach and outlines potential directions for future research.

Reviewers' comments:

Reviewer's Responses to Questions

**Comments to the Author**

1. Is the manuscript technically sound, and do the data support the conclusions?

Reviewer #1: Yes

Reviewer #2: Yes

2. Has the statistical analysis been performed appropriately and rigorously? 

Reviewer #1: Yes

Reviewer #2: Yes

3. Have the authors made all data underlying the findings in their manuscript fully available?

Reviewer #1: Yes

Reviewer #2: Yes

4. Is the manuscript presented in an intelligible fashion and written in standard English?

Reviewer #1: Yes

Reviewer #2: Yes

5. Review Comments to the Author

Reviewer #1: Manuscript Review: "An alternative to chasing an elusive alpha that minimizes ill-defined error costs: simultaneously test a hypothesis at multiple alpha levels"

Overview

The manuscript presents a novel approach to statistical hypothesis testing by proposing the simultaneous testing of a hypothesis at multiple alpha levels within the conventional Neyman-Pearson framework. This method is posited as an alternative to the traditional pursuit of optimal alpha levels, which often involves grappling with ill-defined error costs and assumptions about hypothesis prevalence.

Strengths

Innovative Approach: The manuscript introduces an innovative method to address the long-standing debate about appropriate alpha levels in statistical hypothesis testing.

Addressing Prevalence Sensitivity: It effectively highlights how the optimization of alpha levels is sensitive to assumptions about the prevalence of true hypotheses and the estimated error costs.

Practical Implications: By proposing a multi-level testing approach, the paper suggests a practical solution that can deliver acceptable expected total error costs without the need for precise cost estimations and prevalence assumptions.

Extended Hypotheses Methodology: The approach of extending the parameter space and data with extended hypotheses to overcome logical inconsistencies in multi-level testing is commendable.

Applicability: The paper illustrates its concepts using real-world examples, such as the cost analysis of Molnupiravir therapy in COVID-19, making it relevant and applicable to current research scenarios.

Weaknesses

Complexity and Practicality: The multi-level testing approach, while innovative, may add complexity to the statistical design and analysis process, potentially hindering its practical adoption.

Need for Further Empirical Validation: The manuscript could benefit from more empirical studies or simulations to demonstrate the efficacy and practicality of the proposed method in diverse research scenarios.

Potential for Misinterpretation: Reporting results at multiple alpha levels might lead to confusion or misinterpretation among researchers less familiar with this approach.

Recommendations

Empirical Studies: Conduct empirical studies or simulations to demonstrate the practical implementation and advantages of the proposed method in various research contexts.

Clarity in Reporting: Provide clear guidelines on how to report and interpret results from multi-level testing to avoid potential confusion or misinterpretation.

Consider alternative approaches: Maier and Lakens (2022) also suggest a Bayesian-frequentist compromise to circumvent the problem of having to deal with ill-defined error costs, assumptions of hypothesis prevalence and power estimation. This approach was recently expanded to regression type models by Wulff and Taylor (2023). The paper could benefit from a discussion comparing the proposed approach to this Bayesian-frequentist approach.

Conclusion

The manuscript offers a significant contribution to the field of statistical hypothesis testing by proposing an alternative approach to handling the complexities associated with determining optimal alpha levels. Its practical implications for research design and error cost management are notable. However, the approach's complexity and need for further validation are areas that require attention. The manuscript is a valuable addition to statistical literature and has the potential to influence future research practices significantly.

References

Wulff, J. N., Taylor, L., “How and why alpha should depend on sample size: A Bayesian-frequentist compromise for significance testing”. In: Strategic Organization (2023), in-press. https://journals.sagepub.com/doi/10.1177/14761270231214429

Reviewer #2: The paper “An alternative to chasing an elusive alpha that minimizes ill-defined error costs: simultaneously test a hypothesis at multiple alpha levels” proposes to test at multiple alpha levels simultaneously by assuming different decisions based on which alpha level interval the test is significant at. It shows that testing multiple alphas will often lead to lower expected total costs than testing only at a single alpha. I appreciate the paper's innovative approach of considering multiple alpha levels simultaneously, and the paper shows that this approach can have clear benefits. I think it would make a useful contribution to the literature pending some revisions.

Major comments

- I think one point that is not sufficiently addressed is that interpreting p-values as strength of evidence if not part of the classical Neyman-Pearson view. The author acknowledges that on page 3, but then only gives a very brief description of how Aisbett (2023) overcomes this problem. I think this section would benefit from a more detailed explanation of how the proposed approach relates to significance testing as thought of by Neyman and Pearson.

- Often it is difficult to envision what the multiple decisions based on the different alpha levels would be. Scientists may often decide to either make a claim or not, or in practice, policymakers may decide to either roll out a new drug or not. I think the paper would benefit from more concrete examples that describe the different decisions that would be made as a function of the alpha interval within which the test was significant.

Minor comments

- The paper starts out very theoretical and only gives examples towards the end of the text. I think it would improve readability, especially for experimental researchers, who are the ultimate target audience a lot if the concepts were explained alongside examples rather than putting all the theory first and examples in the second part.

- Paragraph two in the introduction suggests that the probability of the hypothesis being true plays no role in Mudge et al or Maier & Lakens; however, this information is actually also needed to optimise the weighted error rates.

- In Equation 3 a bracket seems to be missing

- The title suggests that the paper is in strong opposition to previous work on optimizing alpha levels; however, when reading the paper, it becomes clear that it is mostly an extension of previous ideas to the multiple alphas and distributions of effect sizes case. I would probably focus the title more on this element (e.g., something like “Don’t put all our eggs in one basket”).

6. PLOS authors have the option to publish the peer review history of their article (what does this mean?). If published, this will include your full peer review and any attached files.

Reviewer #1: No

Reviewer #2: No

---

## [Author Response · Author response to Decision Letter 0]

25 Feb 2024

Reviewer 1 comments and author responses

COMMENT: Conduct empirical studies or simulations to demonstrate the practical implementation and advantages of the proposed method in various research contexts.

RESPONSE: The Discussion now suggests empirical studies in various research contexts as necessary future work. (This follows the advice of the Associate Editor on how to address the comment. Colleagues and I are currently working on applying the approach to studies into performance enhancement in sports science and to the management of invasive plant species.)

COMMENT: Clarity in reporting: Provide clear guidelines on how to report and interpret results from multi-level testing to avoid potential confusion or misinterpretation.

RESPONSE: A new section in Supplement 1 covers reporting of multi-level tests, including confidence intervals and compact letter displays, with supporting R code. (It was hard to see where to address this important recommendation in the main paper without losing the focus on error costs.) 

COMMENT: Consider alternative approaches: Maier and Lakens (2022) also suggest a Bayesian-frequentist compromise to circumvent the problem of having to deal with ill-defined error costs, assumptions of hypothesis prevalence and power estimation. This approach was recently expanded to regression type models by Wulff and Taylor (2023). The paper could benefit from a discussion comparing the proposed approach to this Bayesian-frequentist approach.

RESPONSE: The Discussion now suggests setting multiple alpha levels within this Bayesian-inspired framework, using the associated R code. The idea is to take boundaries of the relative evidence categories as default BFs, which correspond to “default” alphas that are functions of sample size. Relative costs of errors would still have to be considered if setting BFs rather than using defaults. (Thanks for the Wulff & Taylor reference.)

Reviewer 2 comments and author responses.

COMMENT: I think one point that is not sufficiently addressed is that interpreting p-values as strength of evidence if not part of the classical Neyman-Pearson view. The author acknowledges that on page 3, but then only gives a very brief description of how Aisbett (2023) overcomes this problem. I think this section would benefit from a more detailed explanation of how the proposed approach relates to significance testing as thought of by Neyman and Pearson.

RESPONSE: This comment has been addressed in two ways. Firstly, the explanation of the construction in the main paper has been expanded, and an example has been added to Supplement 1. This hopefully clarifies the treatment of the multi-level test as a multiple hypothesis testing problem sitting within the Neyman-Pearson framework. Secondly, findings are now said to be explicitly tied to “test level” instead of “evidence level”. 

COMMENT: Often it is difficult to envision what the multiple decisions based on the different alpha levels would be. Scientists may often decide to either make a claim or not, or in practice, policymakers may decide to either roll out a new drug or not. I think the paper would benefit from more concrete examples that describe the different decisions that would be made as a function of the alpha interval within which the test was significant.

RESPONSE: An example from veterinary science is now used throughout Section 2 to illustrate the various concepts. These examples include multiple decisions, their associated costs, and how costs should link to the multiple test alpha levels. 

COMMENT: The paper starts out very theoretical and only gives examples towards the end of the text. I think it would improve readability, especially for experimental researchers, who are the ultimate target audience a lot if the concepts were explained alongside examples rather than putting all the theory first and examples in the second part.

RESPONSE: This excellent suggestion has been addressed using the veterinary science example—chosen because, regardless of their discipline, many researchers have a personal interest in pets and pet food.

COMMENT: Paragraph two in the introduction suggests that the probability of the hypothesis being true plays no role in Mudge et al or Maier & Lakens; however, this information is actually also needed to optimise the weighted error rates.

RESPONSE: The references have been moved to after the second sentence to make it clearer they refer to both local and global definitions. (Although the global definition in Mudge et al that Maier & Lakens cite has to be extracted from the supplementary R code.)

COMMENT: In Equation 3 a bracket seems to be missing.

RESPONSE: Corrected, thank you.

COMMENT: The title suggests that the paper is in strong opposition to previous work on optimizing alpha levels; however, when reading the paper, it becomes clear that it is mostly an extension of previous ideas to the multiple alphas and distributions of effect sizes case. I would probably focus the title more on this element (e.g., something like “Don’t put all our eggs in one basket”).

RESPONSE: Revised title: “Can expected error costs justify testing a hypothesis at multiple alpha levels rather than searching for an elusive optimal alpha?”

---

## [Decision Letter · Decision Letter 1]

26 Apr 2024

PONE-D-23-36250R1Can expected error costs justify testing a hypothesis at multiple alpha levels rather than searching for an elusive optimal alpha?PLOS ONE

Dear Dr. Aisbett,

Thank you for submitting your manuscript to PLOS ONE. After careful consideration, we feel that it has merit but does not fully meet PLOS ONE’s publication criteria as it currently stands. Therefore, we invite you to submit a revised version of the manuscript that addresses the points raised during the review process.

We have received reviews from two experts in the field—one who participated in the first round of reviews and another who did not. You will notice that Reviewer No. 3 has provided specific guidance on how to enhance your paper. I encourage you to incorporate their feedback on the manuscript into your revision. Specifically, please address the misunderstandings or formal mistakes highlighted in comments 1 and 2. Additionally, points 3 and 4 could enrich the discussion sections or limitations of the paper. As pointed out in point 5, I kindly ask that you introduce a description of the main mechanism earlier in the text. Thank you for attending to these details to further refine your work!

We look forward to receiving your revised manuscript.

Kind regards,

Stephan Leitner

Academic Editor

PLOS ONE

Journal Requirements:

Additional Editor Comments:

We have received reviews from two experts in the field—one who participated in the first round of reviews and another who did not. You will notice that Reviewer No. 3 has provided specific guidance on how to enhance your paper. I encourage you to incorporate their feedback on the manuscript into your revision. Specifically, please address the misunderstandings or formal mistakes highlighted in comments 1 and 2. Additionally, points 3 and 4 could enrich the discussion sections or limitations of the paper. As pointed out in point 5, I kindly ask that you introduce a description of the main mechanism earlier in the text. Thank you for attending to these details to further refine your work!

Reviewers' comments:

Reviewer's Responses to Questions

**Comments to the Author**

1. If the authors have adequately addressed your comments raised in a previous round of review and you feel that this manuscript is now acceptable for publication, you may indicate that here to bypass the “Comments to the Author” section, enter your conflict of interest statement in the “Confidential to Editor” section, and submit your "Accept" recommendation.

Reviewer #1: All comments have been addressed

Reviewer #3: All comments have been addressed

2. Is the manuscript technically sound, and do the data support the conclusions?

Reviewer #1: (No Response)

Reviewer #3: Partly

3. Has the statistical analysis been performed appropriately and rigorously? 

Reviewer #1: (No Response)

Reviewer #3: Yes

4. Have the authors made all data underlying the findings in their manuscript fully available?

Reviewer #1: (No Response)

Reviewer #3: Yes

5. Is the manuscript presented in an intelligible fashion and written in standard English?

Reviewer #1: (No Response)

Reviewer #3: Yes

6. Review Comments to the Author

Reviewer #1: (No Response)

Reviewer #3: The article brings together a host of good ideas that are described and analyzed in sufficient depth. The authors tackle the important problem of linking decision strategies in hypothesis testing to real decision outcomes. Furthermore, the idea of using multiple alpha values when there are multiple decision options seems novel and intriguing to me. The reviewer comments from the previous round of reviews seem mostly addressed to me. Unfortunately, there are multiple mistakes in the notation and equations that took me quite some time to understand. In addition, I have multiple new conceptual points of critique.

General comments:

1. Formal mistakes:

a. In line 190, R – E is introduced as the subset of “true” effects, but E is the subset of the effect sizes for true hypotheses and R the full domain. The term “test hypothesis” being synonymous to the null hypothesis is not a convention in some fields of research (e.g. Psychology). This should be better clarified. In the next paragraph the authors write “… when true effects are further away from R – E, …”, which seems to contradict the previous statements. It would imply that true effects are somehow “further away” from true effects. I suspect that the authors have made at least one accidental flip somewhere in the wording, or, if there is no mistake, need to rewrite the paragraph for understandability.

b. In Equation 3, it is unclear what values the index m of the sum should be able to take. This should be spelled out in the notation. Using all the values given before (0, 1, 2, …, k in lines 215f) does not work, because then C_0(m-1) would not exist for m = 0 and \\alpha_0 is undefined. Furthermore, the second equality cannot be reached by rearrangement, as the authors claimed, because the final closing brackets are misplaced (they belong in front of the \\alpha_m). The Equation seems correct when assuming that m can take the values 1, 2, …, k, moving the bracket, and taking the additional assumptions into account. Even then, the transformation from the left side to the right side are non-trivial (to me). For this reason, it might be worthwhile to add a derivation in the appendix or supplement, or at least expand the explanation of the equation. This erroneous equation has cost me a disproportionate amount of reviewing time.

c. Supplement 2: In Equation S2.1 the variable “d” is included without proper introduction. The effect size notation in the main text suggests to me that “e” would be correct here because d is only used for specific effect sizes in the examples (e.g. Eq. 10). In Equation S2.6 “p” could be introduced more approachably.

2. Individual p-values can be highly unreliable indicators of the strength of the evidence (see e.g., Fig. 1 in Dienes (2014)). On the one hand, the authors provide a mechanism of caution by using multiple alpha-levels. On the other hand, they only implicitly address the issue that there is not only uncertainty that is quantified by the p-value, but also uncertainty in the random draw of the p-value itself. In my view, this should be discussed as a limitation explicitly. It seems especially relevant in one-off applications, in which researchers might develop a false confidence in the exact p-value when using the suggested approach. Only when considering the average error cost across many repetitions does this become a negligible detail that is implicitly treated in the calculations by the authors.

3. Despite discussing decision making in a state of uncertain evidence, the authors do not consider collecting more data as a choice alternative. Maybe this is more obvious in the Bayesian framework, where there are Bayes-factors that are understood as inconclusive and additional data can be collected at will to arrive at a more conclusive result. Within the logic of the current manuscript, the state of inconclusiveness is also explicitly discussed, with partial investment as a suggested typical strategy (e.g. in lines 243 ff.). Given that the authors already consider a palette of choices based on the clarity of the evidence, collecting more data should be discussed as an obvious alternative to minimize error costs. This seems highly relevant if data collection is comparatively cheap. (Remark: The parallel to Bayesian hypothesis testing may generally be interesting to explore. There, researchers may automatically tend towards collecting more data when the Bayes factor is very inconclusive regarding the hypotheses.)

4. A key limitation of the authors’ approach to decision making seems to be that the computation of the cost of errors in empirical hypothesis testing at the time of hypothesis testing may be impossible in many areas. For this reason, it may also be impossible to generate a sensible set of decision alternatives and alpha levels. In other situations, it may not be clear what the potential decisions are. The value of foundational research, for example, lies largely in its role as groundwork for future research and innovation, which is, by definition, unknown at the time. To some degree all research has this knowledge-generating value. More generally it is unclear how (or if) the idea of a set of decisions representing various levels of investment can be applied to general research in the empirical sciences (see also related comment by Reviewer 2 from the first round of reviews). For example, it is unclear why several alpha levels would be needed in a study that has no specific decisions bound to it, yet, and could instead just report the p-value. The authors could more clearly outline under which conditions their approach to hypothesis testing is practical and under which conditions it is rather infeasible or may even be misleading.

a. Related to that (lines 378ff): Operationalizing the costs of non-treatment by only looking at hospitalization without ICU seems to oversimplify the scenario in a dangerous way. This is besides the main point of the study, but maybe an indicator of how difficult it is to determine the costs of decision errors.

5. Only beginning in line 238, the authors clarify that the multiple hypothesis tests are linked to different decisions, and what these decisions could be. This may not be obvious to anyone who first reads this manuscript. Second, they do not state where the intended reduction in error costs comes from in the introduction. In essence, the mechanism behind their approach remains unnamed and undescribed until it is introduced slowly in technical detail. In general, a brief description of the main mechanism and key idea is missing. The introduction could be more straightforward towards these key points to summarize the key idea. Similarly, the beginning of the discussion could focus on the main takeaways earlier instead of first inserting a recollection of the findings by Neves et al. (2020). For example, what I would consider to be one of the main takeaways (lines 543f) is hidden at the end of a paragraph that starts with a different topic.

Specific comments:

line 53: There is a typo in the reference resulting in an impossible publication year.

line 108ff: The term “multi-level” might mislead readers to confuse it with multi-level models, especially if the terminology is widely adopted in future literature or even applied to multi-level models.

line 299: The reader may get the (wrong) impression that the authors imply that a self-censoring version of the publication bias is a non-problematic, standard decision. In any case, the decision to selective not publish null results produces a bias in the literature and therefore has a cost. This topic could be easily avoided since it adds little to the content of the manuscript.

Lines 312ff: The surprisal value of a p-value and the value of the information gained in practice also depend on prior knowledge. The authors seem to assume that no prior knowledge exists and should make this explicit. That the surprisal value given by Rafi & Greenland (2020) is a good general approximation of error costs of real decisions should be justified more clearly. (Remark: At this point researchers might seriously consider the Bayesian framework as an alternative approach, due to the seamless integration of prior knowledge in prior distributions).

Lines 348f: It could be stated more clearly (here and/or above) that the direct relationship between the error costs and the p-value is an explicit simplification for the design of the hypothetical example study, removing the key step of researchers choosing decision alternatives.

Lines 580ff: The authors outline a strategy to arrive at optimal alpha levels based on assumptions about costs of different decisions. In this context it seems highly relevant to me that, all else being equal (including sample size), the p-value has a (non-linear, but) deterministic relationship to the standardized effect size estimate. This is true at least for some simple scenarios, like t-tests. The p-value is a transformation of Cohen’s d in that case. Therefore, any strategy that involves alpha levels implicitly accounts for effect sizes. This might be a relevant transformation for the computation of the costs of decision strategies because they can more easily be linked to standardized effect sizes than p-values given a sample size. In general, I found this worthwhile to remind myself of. It also means that the effect size is implicitly accounted for when using the proposed approach, although it might seem neglected at first glance. Therefore, it seems important to me to point this out somewhere in the manuscript.

Supplement 3: The equation numbers from the main text are shown as “Error! Reference source not found.” In Figure 3, the y-axes are not labeled.

Supplement 4: In Figure S4.1, the y-axes are not labeled.

References:

Dienes, Z. (2014). Using Bayes to get the most out of non-significant results. Frontiers in psychology, 5, 85883.

7. PLOS authors have the option to publish the peer review history of their article (what does this mean?). If published, this will include your full peer review and any attached files.

Reviewer #1: No

Reviewer #3: **Yes: **Nils Petras

---

## [Author Response · Author response to Decision Letter 1]

5 May 2024

Thank you for nominating a third referee who has again provided valuable feedback. My responses to each suggestion are detailed below.

Following this referee’s concern at possible confusion with multilevel models, the term “multi-level” has been replaced throughout the manuscript by “multi-alpha”. This new nomenclature is also used in the responses below.

COMMENT 1a

a. In line 190, R – E is introduced as the subset of “true” effects, but E is the subset of the effect sizes for true hypotheses and R the full domain. The term “test hypothesis” being synonymous to the null hypothesis is not a convention in some fields of research (e.g. Psychology). This should be better clarified. In the next paragraph the authors write “… when true effects are further away from R – E, …”, which seems to contradict the previous statements. It would imply that true effects are somehow “further away” from true effects. I suspect that the authors have made at least one accidental flip somewhere in the wording, or, if there is no mistake, need to rewrite the paragraph for understandability.

RESPONSE TO COMMENT 1a

R – E is now defined less ambiguously as the subset of “effects for which the research hypothesis is true”.

The term “test hypothesis” is now explained when it is first introduced (paragraph 3 of the Introduction) as the “hypothesis that the researcher hopes to reject—typically the null hypothesis of no effect.” It is defined again when specifying the subset E as effects for which the test hypothesis is true. 

The dog diet example is now used to illustrate what it means to be “further away” from the set R – E, with a negative weight loss (dogs on the diet actually put on weight) being distant from the set of losses greater than the smallest meaningful loss 0.1. 

COMMENT 1b

 In Equation 3, it is unclear what values the index m of the sum should be able to take. This should be spelled out in the notation. Using all the values given before (0, 1, 2, …, k in lines 215f) does not work, because then C_0(m-1) would not exist for m = 0 and \\alpha_0 is undefined. Furthermore, the second equality cannot be reached by rearrangement, as the authors claimed, because the final closing brackets are misplaced (they belong in front of the \\alpha_m). The Equation seems correct when assuming that m can take the values 1, 2, …, k, moving the bracket, and taking the additional assumptions into account. Even then, the transformation from the left side to the right side are non-trivial (to me). For this reason, it might be worthwhile to add a derivation in the appendix or supplement, or at least expand the explanation of the equation. This erroneous equation has cost me a disproportionate amount of reviewing time.

RESPONSE TO COMMENT 1b

Apologies for the incorrect bracket positioning in eqn. 3. The range has been added to both summations. In addition, the intermediary steps that follow from the definitions C_0 (0)=0,α_(k+1)=0 are given after the equation, i.e., ∑_(m=1,…,k) (C_0 (m) α_(m+1))=∑_(m=1,…,k-1)(C_0 (m) α_(m+1))≡∑_(m=2,…,k)(C_0 (m-1) α_m)∑_(m=1,…,k)(C_0 (m-1) α_m).

COMMENT 1c. 

Supplement 2: In Equation S2.1 the variable “d” is included without proper introduction. The effect size notation in the main text suggests to me that “e” would be correct here because d is only used for specific effect sizes in the examples (e.g. Eq. 10). In Equation S2.6 “p” could be introduced more approachably.

RESPONSE TO COMMENT 1c

 d is now defined and the reader is referred to the main paper for the definitions of other variables appearing in (S2.1). p is now defined as a probability density function and the expected total error cost ϖ(p,α_m ) is defined in a new (S2.5). 

COMMENT 2

Individual p-values can be highly unreliable indicators of the strength of the evidence (see e.g., Fig. 1 in Dienes (2014)). On the one hand, the authors provide a mechanism of caution by using multiple alpha-levels. On the other hand, they only implicitly address the issue that there is not only uncertainty that is quantified by the p-value, but also uncertainty in the random draw of the p-value itself. In my view, this should be discussed as a limitation explicitly. It seems especially relevant in one-off applications, in which researchers might develop a false confidence in the exact p-value when using the suggested approach. Only when considering the average error cost across many repetitions does this become a negligible detail that is implicitly treated in the calculations by the authors. 

RESPONSE TO COMMENT 2

The following cautionary paragraph has been inserted before the last paragraph of the Discussion: 

"Whatever levels are chosen, testing at multiple alphas is subject to the qualifications needed for any application of P-values, which only indicate the degree to which data are unusual if the test hypothesis together with all other modelling assumptions are correct. Data from a repeat of the trial may still, with the same assumptions and tests, yield a different conclusion." 

COMMENT 3

Despite discussing decision making in a state of uncertain evidence, the authors do not consider collecting more data as a choice alternative. Maybe this is more obvious in the Bayesian framework, where there are Bayes-factors that are understood as inconclusive and additional data can be collected at will to arrive at a more conclusive result. Within the logic of the current manuscript, the state of inconclusiveness is also explicitly discussed, with partial investment as a suggested typical strategy (e.g. in lines 243 ff.). Given that the authors already consider a palette of choices based on the clarity of the evidence, collecting more data should be discussed as an obvious alternative to minimize error costs. This seems highly relevant if data collection is comparatively cheap. (Remark: The parallel to Bayesian hypothesis testing may generally be interesting to explore. There, researchers may automatically tend towards collecting more data when the Bayes factor is very inconclusive regarding the hypotheses.)

RESPONSE TO COMMENT 3

The excellent suggestion of collecting more data replaces the decision to “not publish” that the reviewer pointed out was both unnecessary and controversial in their minor comment line 299. 

COMMENT 4

A key limitation of the authors’ approach to decision making seems to be that the computation of the cost of errors in empirical hypothesis testing at the time of hypothesis testing may be impossible in many areas. For this reason, it may also be impossible to generate a sensible set of decision alternatives and alpha levels. In other situations, it may not be clear what the potential decisions are. The value of foundational research, for example, lies largely in its role as groundwork for future research and innovation, which is, by definition, unknown at the time. To some degree all research has this knowledge-generating value. More generally it is unclear how (or if) the idea of a set of decisions representing various levels of investment can be applied to general research in the empirical sciences (see also related comment by Reviewer 2 from the first round of reviews). For example, it is unclear why several alpha levels would be needed in a study that has no specific decisions bound to it, yet, and could instead just report the p-value. The authors could more clearly outline under which conditions their approach to hypothesis testing is practical and under which conditions it is rather infeasible or may even be misleading.

RESPONSE TO COMMENT 4 (See also the Response to 5).

The difficulty of determining costs is agreed! The examples and simulations aimed to show how expected total costs from multi-alpha tests at pre-set alphas were acceptable if not optimal. The Discussion may have muddied the waters by describing two strategies for choosing alphas that require estimating some costs and decisions. 

The first strategy of taking high and low “optimal” alphas calculated using a range of prevalence models has been expanded to allow for a range of cost models. 

The second strategy based on stakeholders determining decisions and costs is now explicitly said to be “appropriate to applied research.”

The third strategy of using default alpha values does not depend on estimating costs and has been highlighted by:

 (i) expanding the conclusion of the Abstract as follows: 

"Testing at multiple default thresholds removes the need to formally identify decisions, or to model costs and prevalence as required in optimization approaches. Although total expected error costs with this approach will not be optimal, our results suggest they may be lower, on average, than when “optimal” test levels are based on mis-specified models."

 (ii) reorganizing the Discussion so that it is the last of the three strategies to be presented.

The Introduction pointed out that an advantage of testing at multiple levels over simply reporting P-values is the a priori attention given to the relative importance of Type I and Type II errors, and to sample size. That is, even though every researcher should report raw P-values, at the design stage of any study it is essential to think about whether the study has the power to detect potential effect sizes. I think this applies broadly. 

COMMENT 4a

Operationalizing the costs of non-treatment by only looking at hospitalization without ICU seems to oversimplify the scenario in a dangerous way. This is besides the main point of the study, but maybe an indicator of how difficult it is to determine the costs of decision errors. 

RESPONSE TO COMMENT 4a

The source paper on Covid treatment costs had data on mechanical ventilation as another layer above ICU, so a fuller analysis would have made the example unnecessarily complicated. That said, it is true that estimating costs is always going to be complicated. 

COMMENT 5

Only beginning in line 238, the authors clarify that the multiple hypothesis tests are linked to different decisions, and what these decisions could be. This may not be obvious to anyone who first reads this manuscript. Second, they do not state where the intended reduction in error costs comes from in the introduction. In essence, the mechanism behind their approach remains unnamed and undescribed until it is introduced slowly in technical detail. In general, a brief description of the main mechanism and key idea is missing. The introduction could be more straightforward towards these key points to summarize the key idea. Similarly, the beginning of the discussion could focus on the main takeaways earlier instead of first inserting a recollection of the findings by Neves et al. (2020). For example, what I would consider to be one of the main takeaways (lines 543f) is hidden at the end of a paragraph that starts with a different topic.

RESPONSE TO COMMENT 5

The Abstract now explains “Error costs are tied to decisions, with different decisions assumed for each of the potential outcomes in the multi-alpha level case.” 

The paragraph in the Introduction concerning losses/payoffs in relation to test levels has been clarified and expanded to read: "Thus, rather than a statistical test leading to dichotomous decisions, there may be a range of decisions, each associated with different error costs. A small P-value may be interpreted as strong evidence that triggers a decision bringing a large positive payoff when the finding is correct and a large loss when the finding is incorrect as compared with the payoff when no evidence level is reported. "

The paragraph near the start of the Discussion on Neves et al (2022) has been removed. 

MINOR COMMENTS

line 53: There is a typo in the reference resulting in an impossible publication year. 

RESPONSE: Changed to 2019.

line 108ff: The term “multi-level” might mislead readers to confuse it with multi-level models, especially if the terminology is widely adopted in future literature or even applied to multi-level models. 

RESPONSE: Changed to “multi-alpha” tests throughout.

line 299: The reader may get the (wrong) impression that the authors imply that a self-censoring version of the publication bias is a non-problematic, standard decision. In any case, the decision to selective not publish null results produces a bias in the literature and therefore has a cost. This topic could be easily avoided since it adds little to the content of the manuscript. 

RESPONSE: Changed as described in response to comment 3.

Lines 312ff: The surprisal value of a p-value and the value of the information gained in practice also depend on prior knowledge. The authors seem to assume that no prior knowledge exists and should make this explicit. That the surprisal value given by Rafi & Greenland (2020) is a good general approximation of error costs of real decisions should be justified more clearly. (Remark: At this point researchers might seriously consider the Bayesian framework as an alternative approach, due to the seamless integration of prior knowledge in prior distributions).

RESPONSE: The subsection linking costs to surprisal values now notes the assumption of no prior knowledge and gives an example of typical prior knowledge. (It is true that a Bayesian approach often makes more sense, but is outside the scope of this manuscript.)

Supplement S2 now shows how the relationship between costs and surprisal values can be used to set alpha levels if costs are known. Conversely, if alphas are pre-set (e.g. are defaults) and decisions concern the extent of rollout of a treatment, as in the dog food example, it shows how the relationship can be used to determine the appropriate decision for a given alpha level. 

Lines 348f: It could be stated more clearly (here and/or above) that the direct relationship between the error costs and the p-value is an explicit simplification for the design of the hypothetical example study, removing the key step of researchers choosing decision alternatives.

RESPONSE: Added: “This simplifying assumption implies no prior knowledge about test outcomes.”

Lines 580ff: The authors outline a strategy to arrive at optimal alpha levels based on assumptions about costs of different decisions. In this context it seems highly relevant to me that, all else being equal (including sample size), the p-value has a (non-linear, but) deterministic relationship to the standardized effect size estimate. This is true at least for some simple scenarios, like t-tests. The p-value is a transformation of Cohen’s d in that case. Therefore, any strategy that involves alpha levels implicitly accounts for effect sizes. This might be a relevant transformation for the computation of the costs of decision strategies because they can more easily be linked to standardized effect sizes than p-values given a sample size. In general, I found this worthwhile to remind myself of. It also means that the effect size is implicitly accounted for when using the proposed approach, although it might seem neglected at first glance. Therefore, it seems important to me to point this out somewhere in the manuscript.

RESPONSE: I think the best potential application of this transformation may be in setting multiple default alpha levels. This sentence has therefore been added to the Discussion. "Default alpha levels might also be assigned indirectly using default effect sizes, such as the Cohen’s d values for “small”, “medium” and “large” effects, given the functional relationships between P-values and effect sizes that depend, inter alia, on sample size and estimated variance."

MISSING AXIS LABELS etc. - RESPONSE HAS BEEN TO FIX

Supplement 3: The equation numbers from the main text are shown as “Error! Reference source not found.” In Figure 3, the y-axes are not labeled.

Supplement 4: In Figure S4.1, the y-axes are not labeled.

NOTE: Also labelled y-axes in Figs 1a and 1c,

---

## [Decision Letter · Decision Letter 2]

16 May 2024

Can expected error costs justify testing a hypothesis at multiple alpha levels rather than searching for an elusive optimal alpha?

PONE-D-23-36250R2

Dear Dr. Aisbett,

We’re pleased to inform you that your manuscript has been judged scientifically suitable for publication and will be formally accepted for publication once it meets all outstanding technical requirements.

Kind regards,

Stephan Leitner

Academic Editor

PLOS ONE

Additional Editor Comments (optional):

I am pleased to accept this paper for publication. The reviewer has raised one final minor point which I believe is significant. I recommend that the authors address this comment while preparing their final submission. Congratulations on producing such an excellent piece of work!

Reviewers' comments:

Reviewer's Responses to Questions

**Comments to the Author**

1. If the authors have adequately addressed your comments raised in a previous round of review and you feel that this manuscript is now acceptable for publication, you may indicate that here to bypass the “Comments to the Author” section, enter your conflict of interest statement in the “Confidential to Editor” section, and submit your "Accept" recommendation.

Reviewer #3: All comments have been addressed

2. Is the manuscript technically sound, and do the data support the conclusions?

Reviewer #3: Yes

3. Has the statistical analysis been performed appropriately and rigorously? 

Reviewer #3: Yes

4. Have the authors made all data underlying the findings in their manuscript fully available?

Reviewer #3: Yes

5. Is the manuscript presented in an intelligible fashion and written in standard English?

Reviewer #3: Yes

6. Review Comments to the Author

Reviewer #3: There is one minor detail that I think ought to be corrected in the publication process:

- In lines 309ff the authors introduced additional data collection (“plan a larger study”) as a decision option when the initial study yielded non-significant results. In unbiased science, the collection of additional data serves the purpose of producing clearer evidence, not more significant evidence. In other words, additional data collection may be indicated because the initial study results are inconclusive, not because they are non-significant. Although basic null-hypothesis statistical testing does not allow strong conclusions in favor of the null-hypothesis, it can be modified to conclusively reject an alternative (research) hypothesis by adding a smallest effect size of interest (as the authors alluded to in lines 199ff). Several strategies of collecting additional data contingent on non-significant results (e.g. data-peeking, selective publishing among repeated studies) qualify as p-hacking and are therefore questionable research practices. I doubt that the authors intended to suggest indifference to or even support of such practices. I suggest changing this sentence.

7. PLOS authors have the option to publish the peer review history of their article (what does this mean?). If published, this will include your full peer review and any attached files.

Reviewer #3: **Yes: **Nils Petras

---

## [Editor Report · Acceptance letter]

23 May 2024

PONE-D-23-36250R2 

PLOS ONE

Dear Dr. Aisbett, 

I'm pleased to inform you that your manuscript has been deemed suitable for publication in PLOS ONE. Congratulations! Your manuscript is now being handed over to our production team.

Kind regards, 

on behalf of

Dr. Stephan Leitner 

Academic Editor

PLOS ONE